# Educational technologies for teaching hand hygiene: Systematic review

**Daiane Rubinato Fernandes**[1⊙], **Bruna Nogueira dos Santos**[1⊙], **Carolina Scoqui Guimarães**[1⊙], **Elaine Barros Ferreira**[2‡], **Amanda Salles Margatho**[1‡], **Paula Elaine Diniz dos Reis**[2‡], **Didier Pittet**[3‡], **Renata Cristina de Campos Pereira Silveira**[1‡]*

1 Ribeirão Preto College of Nursing, University of São Paulo, Ribeirão Preto, São Paulo, Brazil, 2 University of Brasília, Brasília, Distrito Federal, Brazil, 3 University of Geneva Hospitals, Genève, Switzerland

⊙ These authors contributed equally to this work.
‡ EBF, ASM, PEDR, DP and RCCPS also contributed equally to this work.
* recris@eerp.usp.br

## Abstract

### Aim

To gather available scientific evidence on technologies used to teach hand hygiene to professional populations and lays involved in health care in the hospital setting. This systematic review was designed as proposed by Preferred Reporting Items for Systematic Reviews and Meta-Analysis, included studies reporting primary, original, quantitative research findings with no date limit and written in English, Spanish or Portuguese. The search was performed in the following electronic databases: Cochrane Library, Cumulative Index to Nursing and Allied Health Literature, *Excerpta Medica dataBASE*, *Literatura Latino-Americana e do Caribe em Ciências da Saúde*, US National Library of Medicine, Scopus, Web of Science, Google Scholar and ProQuest. The eligibility criteria were applied independently by two reviewers to select the studies, first by reading the titles and abstracts on the Rayyan platform and then by full text reading of the eligible studies. After a descriptive analysis, the studies were subjected to critical evaluation of their methodological quality using JBI tools.

### Results

Seven studies were included, addressing various methods for teaching hand hygiene using different technological resources, such as audiovisual electronic devices, videos, virtual reality, and gamification using tablets and smartphones, in different populations.

### Conclusion

Using technologies to teach hand hygiene considerably helps patients, visitors, and relatives in learning the procedures and efficiently improves hand hygiene compliance rates among healthcare professionals, creating evidence-based repetitive learning opportunities for patients and caregivers.

**Data Availability Statement:** All relevant data are within the paper.

**Funding:** This study was financed in part by the Coordenação de Aperfeiçoamento de Pessoal de

Nível Superior – Brasil (CAPES) – Finance Code 001.

**Competing interests:** The authors have declared that no competing interests exist.

## Introduction

Hand hygiene (HH) is globally accepted as a key infection prevention and control strategy and as the most important and least expensive activity to mitigate harm related to healthcare-associated infections (HAIs). Yet, despite its widely established relevance, HH implementation is still a barrier in healthcare settings because HH compliance (HHC) rates remain low among healthcare professionals (HCPs), patients, and relatives. In fact, studies show that this compliance rate rarely exceeds 40% [1, 2].

The prevention of HAIs should be a priority action for healthcare organizations. According to the WHO, out of 100 patients, seven will acquire at least one healthcare-associated infection during their hospital stay, the risk doubles and may even increase 20 times in low- and middle-income countries. The sicker and more fragile patients become, the greater becomes the risk of HAI and their deathly consequences [3].

Infection prevention and control interventions can reduce infections in healthcare by 70%, specially HH programs can reduce the risk by more than half of dying as a result of infections, and also reduce associated long-term complications and healthcare costs [4].

Everyone involved in patient care must clean their hands correctly because the HH procedure becomes inefficient when performed incorrectly [5]. HH should only be deemed successful when the appropriate technique is performed, which requires cleaning all surfaces of the hands, and at all moments recommended by the World Health Organization (WHO) in one of the main documents on this topic, at "Five moments for hand hygiene" [6]. For this purpose, HH teaching strategies have been studied towards identifying those with the strongest positive impact on the improvement in the technique of HH and long-term HHC, given the difficulties in establishing and implementing teaching strategies for promoting permanent or at least long-lasting behavior changes.

By enabling health teams to overcome barriers to repetitive exposure to content [5, 7], educational technologies can be used as a teaching tool to enhance knowledge on HH. Although educational technologies are not the solution to the problem, they are a promising tool to improve HHC. In the literature, educational technologies are described as teaching-learning strategies for accessing, gathering, handing, presenting, or communicating information in an accessible, understandable, and continuous way by making the educational process more appealing and playful and by exemplifying the content. Examples of these tool include virtual reality, smartphones, tablets, electronic games, and educational videos [7, 8].

To this end, technology is a dynamic and permanent teaching platform in which theory is quickly combined with practice, unlike other methods, which are monotonous, discouraging, and repetitive. Thus, this teaching method facilitates the work of HCPs, helping them communicate with family members and patients, and providing care guidelines [9].

However, studies indicate that these technologies are inadequately used because many options currently available are not validated or based on evidence. Furthermore, the dissemination of unreliable content often further hinders the learning process. For this reason, even though HH significantly reduces preventable infections, morbidity, and mortality, there is still some resistance to the use of technology to teach HH.

Considering the above, the present systematic review (SR) aims at gathering scientific evidence related to the use of technologies for teaching professional and lay populations (family members, visitors, relatives, and patients) hand hygiene and at supporting and encouraging their development for this purpose.

| | |
|---|---|
| **Handwashing** | Washing hands with plain or antimicrobial soap and water. |
| **Hand cleansing** | The action of performing hand hygiene for the purpose of physically or mechanically removing dirt, organic material, and/or microorganisms. |
| **Hand hygiene** | Use soup and water, or hand rubbing using an alcohol based handrub formulation. |
| **Hand hygiene compliance** | Washing hands with soap and water or using alcohol-containing preparations before touching a patient before performing any clean/aseptic procedure; after body fluid exposure/risk; after touching a patient; after touching patient surroundings. |
| **Compliance rate** | Calculated by dividing the number of observed hand hygiene 'moments' where proper hand hygiene was practiced by the total number of opportunities for HH and multiplying by 100[4]. |

**Fig 1. Definitions adopted in the SR.**

## Materials and methods

This SR was designed and written as proposed by Preferred Reporting Items for Systematic Reviews and Meta-Analysis (PRISMA) [10]. The protocol was published in the platform Open Science Framework [11].

The definitions adopted in this review are shown in Fig 1.

The research question, formulated using the PICOS strategy, is "What is the evidence on technologies used HH to teach patients, caregivers, and healthcare professionals?" (Fig 2).

The studies were identified using the following electronic databases: Cumulative Index to Nursing and Allied Health Literature (CINAHL), Cochrane Library, *Excerpta Medica* dataBASE (EMBASE), Latin American and Caribbean Health Sciences Literature (LILACS), US National Library of Medicine (PubMed), Scopus database, and Web of Science. In addition, a search for grey literature was performed on Google Scholar and ProQuest Thesis and Dissertations.

To construct the search strategy, the following descriptors were identified in Decs/Mesh, and used: "technology", "health technology" and "hand hygiene" and their terms combined with AND or OR, grouped and adapted according to the specificities of each database.

| Acronym | Definition | Description |
|---|---|---|
| **P** | Population | Technologies on hand hygiene used by patients, caregivers, and healthcare professionals |
| **I** | Intervention | Technology application method |
| **C** | Comparison | Not applied |
| **O** | Outcome | Technology usability |
| **S** | Study type | Primary studies, with a quantitative research approach, including randomized and non-randomized clinical trials, quasi-experimental studies, cohort, case-control, and cross-sectional studies. |

**Fig 2. PICOS strategy used to formulate the research question.**

The search strategy was developed by professionals in the area of evidence synthesis and infection prevention and control, and technically evaluated by a librarian. Subsequently, tests were performed on the target databases to assess whether they were sensitive to the research question. The tests and detailed terms of the search strategy are shown in Fig 3.

The search in electronic databases was performed on January 6th 2022, and updated on May 23rd 2023.

The results were exported to the EndNote Basic [12] to remove duplicated references and then imported to the platform Rayyan [13] to select the articles.

In Rayyan, two reviewers (D.R.F. and F.D.B.) independently selected the studies, by reading the title and the abstract with the blinding tool on, for a reliable and unbiased selection process based on the eligibility criteria. Studies deemed eligible were then analyzed by the reviewers, who read the articles in full. Upon divergences between reviewers, a third reviewer (R.C.C.P.S.) with expertise on the theme was consulted, albeit without accessing the full selection process to ensure an impartial decision.

This review included studies reporting primary, quantitative research findings identifying the use of educational technologies about HH among populations of health professionals, patients, family members and visitors involved in health care in the hospital setting, of any age group, published in English, Portuguese or Spanish, no data limit. Excluded from this review were case reports, case series, secondary studies (other reviews), editorials, letters to the editor, books, book chapters, guidelines, expert opinion articles, experience reports, conference proceedings and abstracts, dissertations and theses, in addition to studies outside the scope of this review.

Data from the studies were collected using a template, which includes the following information: reference, year of publication and country of the study, objective, study design, sample size, technologies used to teach HH, and main outcomes, and were qualitatively analyzed after gathering descriptive evidence.

The methodological quality of the primary research studies included in the sample was assessed using JBI critical appraisal tools [14]. Independently, two reviewers (D.R.F. and L.G.V.) evaluated the studies included in the sample using the appropriate appraisal tool for each study design. The possible answers as to where the studies met the methodological quality criteria of this SR were "yes," "unclear," "no," or "not applicable". The third reviewer (R.C.C.P.S.) was required to resolve possible conflicts in this critical appraisal.

No conflicts of interest compromised the analysis of the results from this SR. Since the review did not involve human participants, it did not require ethical committee approval. However, the authors conducted this review with a strong emphasis on transparency and reproducibility, avoiding bias, and ensuring a comprehensive and fair evaluation.

## Results

In total, 701 studies were identified in the databases, of which 74 were duplicates and therefore excluded from the sample. As a result, 627 studies were screened by reading the title and abstract. Among them, 573 were excluded for failing to meet the eligibility criteria of this review. Thus, 54 studies were selected for full-text reading (Fig 4).

After full-text reading, 47 articles were excluded, among which 13 (27.6%) were conference proceedings, 28 (59.6%) did not answer the research question, two (4.3%) were editorials, two (4.3%) reported secondary research findings, one (2.1%) was a case report, and one (2.1%) was not found.

At the end of this process, seven studies were considered eligible for this SR. Lastly, the references of the studies deemed eligible, and the grey literature were searched, but no potentially

| Database | Search Strategy |
|---|---|
| PubMed | (((("caregivers"[Mesh:NoExp] OR "caregivers"[All Fields] OR caregiver* OR carer* OR "care givers" OR "care giver" OR "family caregivers" OR "family caregiver" OR "parents"[MeSH Terms] OR "parents"[All Fields] OR "Patients" [Mesh:NoExp] OR "Health Care Workers"[All Fields] OR "health personnel"))) AND (("Technology"[Mesh] OR "Technology"[All Fields] "Educational Technology" [MeSH] OR "Educational Technology" [All Fields] OR "Educational Technologies" OR "Instructional Technology" OR "Instructional Technologies" OR "Biomedical Technology"[Mesh] OR "Biomedical Technology"[All Fields] OR "Biomedical Technologies"))) AND (("hand hygiene"[MeSH Terms] OR "hand hygiene"[All Fields] OR "hand disinfection"[MeSH Terms] OR "hand disinfection"[All Fields] OR "hand sanitization" OR "handwashing" OR "hand washing" OR "hand washings")) |
| Scopus | "hand hygiene" OR "hand disinfection" OR "hand sanitization" OR "handwashing" AND "Educational Technology" OR "Educational Technologies" OR "Instructional Technology" OR "Instructional Technologies" OR "Biomedical Technology" OR "Biomedical Technologies" AND "caregivers" OR caregiver OR "care givers" OR "care giver" OR "family caregivers" OR "family caregiver" OR parents OR patients OR "health personnel" OR "health care workers" |
| Embase | ('caregivers' OR 'caregiver' OR 'care givers' OR 'care giver' OR 'family caregivers' OR 'family caregiver' OR 'parents' OR 'patients' OR 'health personnel' OR 'health care workers') AND ('technology' OR 'educational technology' OR 'educational technologies' OR 'instructional technology' OR 'instructional technologies' OR 'biomedical technology' OR 'biomedical technologies') AND ('hand hygiene' OR 'hand disinfection' OR 'hand sanitization' OR 'handwashing' OR 'hand washing' OR 'hand washings') |
| CINAHL | (("caregivers" OR caregiver OR "care givers" OR "care giver" OR "family caregivers" OR "family caregiver" OR "parents" OR "Patients" OR "Health Personnel" OR "health care workers")) AND (("Technology" OR "Educational Technology" OR "Educational Technologies" OR "Instructional Technology" OR "Instructional Technologies" OR "Biomedical Technology" OR "Biomedical Technologies")) AND (("hand hygiene" OR "hand disinfection" OR "hand sanitization" OR "handwashing" OR "hand washing" OR "hand washings")) |
| Cochrane Library | Trials matching (("caregivers" OR caregiver OR "care givers" OR "care giver" OR "family caregivers" OR "family caregiver" OR "parents" OR "Patients" OR "Health Personnel" OR "health care workers")) AND (("Technology" OR "Educational Technology" OR "Educational Technologies" OR "Instructional Technology" OR "Instructional Technologies" OR "Biomedical Technology" OR "Biomedical Technologies")) AND (("hand hygiene" OR "hand disinfection" OR "hand sanitization" OR "handwashing" OR "hand washing" OR "hand washings")) |
| Web of Science | (("caregivers" OR caregiver OR "care givers" OR "care giver" OR "family caregivers" OR "family caregiver" OR "parents" OR "Patients" OR "Health Personnel" OR "health care workers")) AND (("Technology" OR "Educational Technology" OR "Educational Technologies" OR "Instructional Technology" OR "Instructional Technologies" OR "Biomedical Technology" OR "Biomedical Technologies")) AND (("hand hygiene" OR "hand disinfection" OR "hand sanitization" OR "handwashing" OR "hand washing" OR "hand washings")) |
| LILACS | ("caregivers" OR "parents" OR "Patients" OR "Health Personnel" OR ("cuidadores" OR "pais" OR "Pacientes" OR "Profissionais de saúde" OR ("cuidadores" OR "padres" OR "Pacientes" OR "Personal de salud") ("Technology" OR "Educational Technology" OR "Biomedical Technology" OR "Tecnologia" OR "Tecnologia Educacional" OR "Tecnologia Biomédica" OR "Tecnología" OR "Tecnología educativa" OR "Tecnología biomédica") ("hand hygiene" OR "hand disinfection" OR "higiene das mãos" OR "desinfecção das mãos" OR "higiene de las manos" OR "desinfección de las manos") |

**Fig 3. Search strategy for electronic databases.**

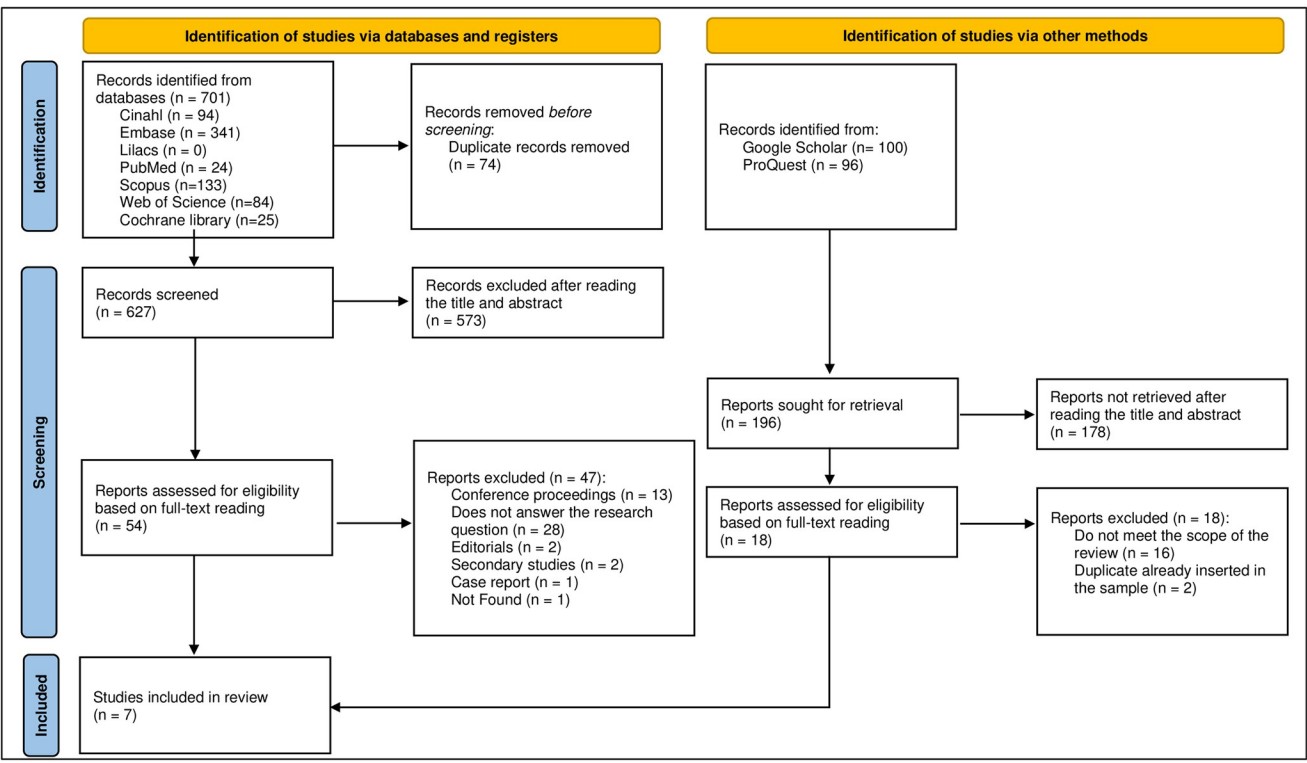

**Fig 4. Flowchart of the selection, inclusion, and exclusion process of studies.**

eligible studies were identified for inclusion in this SR, thus resulting in a final sample of seven primary studies being subjected to synthesis without meta-analysis and critical appraisal of their methodological quality.

Fig 5 shows the synthesis of the studies included in this SR [15–21].

The sample of this SR consisted of three randomized studies (42.8%) [18, 19, 21], three cross-sectional studies (42.8%) [13, 15, 18], and one quasi-experimental study (14.4%) [16].

By country of origin, 28.6% of the studies were conducted in Asian countries: South Korea (n = 1) and Taiwan (n = 1); 42,8% in European countries: United Kingdom (n = 1), Portugal (n = 1), Switzerland (n = 1); and 28.6% in North America: Canada (n = 1) and the United States of America (n = 1). The studies included in this SR were performed in pediatric wards of a university hospital, in intensive care units (ICUs), one of which was a pediatric ICU, in a veteran's hospital, and in a university.

Most studies were published in medical journals (85.7%, n = 6), and only one study was found in a nursing journal (14.3%, n = 1).

All study recruited participants aged 18 years or older, except for one study whose sample consisted of children from 3 to 17 years of age. The populations investigated in the studies included in this SR were predominantly HCPs, such as nurses, physicians, and physical and occupational therapists, with studies on this population comprising 42.8% (n = 3) of the sample in this review. One study was performed with university students 14.3% (n = 1), and the samples of the other studies consisted of patients, family members, and visitors.

These studies addressed different ways of teaching HH using technological resources. As they targeted different populations, different approaches and contents were also needed to teach HH.

| Author,<br>Year, Journal (impact factor),<br>Country, Study design | Objective | Intervention | Main outcomes |
|---|---|---|---|
| Chen and Chian, 2007<br><br>Journal of Clinical Nursing (1.301)<br><br>China<br><br>Quasi-experimental study | To compare families who watched a video designed to demonstrate the handwashing technique with families who learned the same technique using an illustrated poster in terms of compliance with and improvement in handwashing skills. | EG[*], 61 families, was subjected to a video-centered teaching program to demonstrate hand-washing technique. CG[†], 62 families, learned the same techniques using an illustrated poster. A 20-item hand-washing checklist was used to examine hand-washing compliance and accuracy.<br>Total n: 123 families (mothers, grandfathers, grandmothers, aunts and uncles).<br>Data collection period: 5 days. | The patients aged 30 or older. The compliance score increased from 7.0 to 8.6 in EG[*] and from 4.7 to 5.9 in CG[†]. Handwashing compliance scores increased during the following five visits (p < 0.0001). The improvement in handwashing scores was not higher in EG families[*] than in CG[†] families (p= 0.0798). Accuracy scores increased from 3.7 to 5.7 in EG[*] and from 2.7 to 3.7 in CG[†] (p= < 0.0001). The improvement in accuracy scores was significantly higher in EG[*] families than in CG[†] families (p= 0.0003). The accuracy of the handwashing technique significantly increased over time in both groups during the study period. |
| Boscart et al., 2008<br><br>Journal of Hospital Infection - Elsevier (2.956)<br><br>Toronto, Canada<br><br>Exploratory descriptive study | To explore the acceptability and usability of a new HH system created with 3 main components: protected zones (installed to define individual patient environments), alcohol-based hand sanitizer dispensers, and small wearable electronic monitors (placed on the body). | 14 participants each in four groups: the first group consisted of physical and occupational therapists; the second, nurses; the third, doctors; the fourth, professional cleaners. The first phase of the study was a field test in which participants tested the portable device and commented on their experience by filling out a questionnaire. The second phase of the study consisted of a focus group with the same participants to explore the experience of the field test, namely the acceptability of being monitored and the perceived potential usefulness of the device. | The participants were 44.3 years old on average and responded positively to questions about device usability and reminder signal to perform hand cleansing. However, when failing to disinfect the hands, 7 staff would like to have only 1 more reminder, whereas the other 7 would prefer up to 2 reminder signals. Auditory and visual reminders were perceived as potentially disturbing for both HCPs and patients. As for the small wearable electronic monitor, most employees indicated that the device should be lighter. Regarding the preferred location for wearing the device, 3 staff indicated the hip pocket; 3, an arm belt; 3, a lanyard; and 3, a belt. Participants were comfortable receiving information about individual performance and were also interested in anonymous group information for comparison purposes. |
| Lary et al., 2020<br><br>Journal of Infection Prevention (0.0)<br><br>United Kingdom<br><br>Cluster randomized study | To test whether two interactive educational interventions improve children's and their visitors' HHC[‡] | The educational intervention for HHC[‡] was either using the Glo-yo (ultraviolet light) (demonstrated during a 30-minute session) or a video on a mobile phone (mobile learning technology (MLT[§])), demonstrated during a 30-min session by a research student).<br>Total n: 31 participants (16 visitors and 15 patients) | The children were between 3 and 17 years old. The improvement in HHC[‡] was great er after the intervention session with Glo-yo (p <0.001), particularly among children, with an increase from 0 to 61.5% in HHC[‡]. Among the wards who received the MLT[§] intervention, HHC[‡] remained similar to that of the baseline for children (39.5 and 37.5%, respectively). With the control handout, HHC[‡] increased from 0% at baseline to 26.4% post-intervention among children. Among visitors, HHC[‡] increased by 16, 22.4%, and 34.2% with the control handout, MLT[§], and Glo-yo, respectively. |
| Marques et al., 2017<br><br>Bmc Medical Informatics and Decision Making (2.134)<br><br>Portugal<br><br>Cross-sectional study | To develop a gamification solution (using game elements and design in other contexts) that can provide HCPs with real-time feedback on personal HHC towards raising their awareness of HHC, changing their behaviors, and optimizing their performance | The study adopted the Design Science Research Methodology (DSRM[**]), which consisted of six activities, but reported the methods and results of only three activities (Design, Development, and Evaluation). Preliminary experiments, simulations, and field studies were performed in an ICU[††] of a Portuguese tertiary referral hospital. The ICU[††] nurses were included in a focus group during the research, participating in various sessions throughout the implementation process.<br>N: 6 nurses.<br>Data collection period: 10 months. | Nurses aged between 25 and 60 years liked the concept because it provides a unique opportunity to receive feedback on their performance. In the first work iteration, tests performed with indoor location technology applied for distance estimation showed an unacceptable lack of accuracy. In proximity-based technology, the sequence of positions could be identified, but the beacons were unstable. In the second work iteration, a different internal localization technology was explored, which did not work properly either, precluding testing the solution as a whole (gamification application included). |
| Knighton et al., 2018<br><br>American Journal of Infection Control - Elsevier (1.971)<br><br>United States<br><br>Cross-sectional study | To assess the effectiveness of adding a new verbal EAR to a patient HH bundle | Some participants (n = 41) received HH bundle 1 (PHHB1), which included a video, a handout, and a personalized verbal EAR[‡‡] that prompted hand cleansing at 3 mealtimes a day. The other participants (n=34) received hand hygiene bundle 2 (PHHB2), which included the identical video and handout, but not the EAR[‡‡].<br>Total n: 75 patients | PHHB1 (mean age: 66.3 years) averaged significantly more use of hand sanitizer product over the 3 days of the study than the PHHB2 (mean age: 65.5 years) (p≤0.001). On day 3, PHHB1 used approximately 19g more of alcohol-based sanitizer than PHHB2. PHHB1 used alcohol-based hand sanitizer approximately 4-5 times a day, whereas PHHB2, on average, used alcohol-based hand sanitizer 1-2 times a day. |
| Choi and Noh, 2020<br><br>Asian Journal of Communication (2,074)<br><br>South Korea<br><br>Cross-sectional study | To explore how exposure to a virtual reality HH promotion video influences users' HH intention | The participants were provided with virtual reality devices and watched a 5-minute 360-degree first-person virtual reality video promoting handwashing. Subsequently, they completed a questionnaire about their virtual reality video experiences, including presence, flow, fear, attitude towards handwashing, and handwashing intention.<br>Total n: 150 students (75 women and 75 men).<br>Data collection period: 2 weeks. | The average age was 21.6 years. Presence in the virtual reality environment was positively correlated with flow, that is, the psychological state that indicates total immersion and focus on a given activity. Flow played a key role in increasing message acceptance level and fear responses to the handwashing promotion video content. Furthermore, the higher the level of acceptance of the handwashing promotional content message, the more favorable the users' attitude towards handwashing. This correlation, in turn, helped to facilitate handwashing intention. |
| Pires et al, 2021<br><br>JAMA Netw Open (13.36)<br><br>Geneva, Switzerland<br><br>Randomized Clinical Trial | To determine whether providing real-time feedback on a simplified HH action improves compliance with the World Health Organization's "5 Moments" and the quality of the HH action. | This open-label, cluster randomized, stepped-wedge clinical trial was conducted between June 1, 2017, and January 6, 2018 (with a follow-up in March 2018), in a geriatric hospital of the University of Geneva Hospitals, Switzerland. All 12 wards and 97 of 306 eligible health care workers (HCWs) volunteered to wear a novel electronic wearable device that delivered real-time feedback on duration of hand rubbing and application of a hand-sized customized volume of alcohol-based handrub.<br><br>Total n: 97 professionals<br>Data Colletion period: between June 1, 2017, and January 6, 2018 | The use of this device did not change adherence to HH, but it increased the duration of hand rubbing and the volume of product used by health professionals. |

**Fig 5. Synthesis of the studies (n = 7) included in the systematic review.** [*]EG: experimental group; [†]CG: comparison group; [‡]HHC: hand hygiene compliance; [§]MLT: mobile learning technology; [**]DSRM: Design Science Research Methodology; [††]ICU: Intensive Care Unit; [‡‡]EAR: personalized verbal electronic audio reminder.

In one study [15], the educational technology used in teaching HH was a wearable electronic device that emitted audiovisual warning signals reminding HCPs to perform HH and provided feedback on the quality of the technique and their performance in a field test. Based on the participants' answers to the questionnaires, acceptability, usability, feedback, and the warning sign were positively regarded by the team. However, the audiovisual signals were regarded as potentially disturbing for patients and HCPs.

In a study [16] on exposure to educational content on HH comparing two groups of families of children in pediatric intensive care units, namely a control group being taught using illustrations as teaching materials and an experimental group being taught using a video presentation as educational technology, the results showed that both interventions improved the HH technique, but the experimental group significantly outperformed the control group ($\beta = 0.02263$, $p = 0.0003$).

In another study [20], the authors used educational technology involving a gamification solution to accurately collect data and provide real-time feedback on nurses' HH. For this purpose, two work iterations were performed by applying gamification components. Each work iteration used a different indoor location technology, and the participants received information through wireless sensors. This information was sent to a web service, and feedback was provided through a gamification application accessed by the nurses via a link. The nurses liked this model of analyzing and receiving feedback on their HH performance. Even though the participants liked this new concept, technical problems occurred in both iterations, limiting the complete analysis of these technologies. From the first to the second work iteration, the technology improved, but even after technical adjustments to the technology, some problems remained; therefore, in this study, the technology system was unstable, and the improvements in the sensors were insufficient to solve the problems.

A randomized study [19] tested educational interventions for improving HHC among children and their visitors. The Glo-yo (ultraviolet light) intervention increased HHC by 61.5% among children and by 34.2% among visitors. In contrast, HHC for the wards receiving a mobile learning technology (MLT) intervention remained similar to that of the baseline phase among children and visitors (39.5 and 37.5%, respectively), and a result that was similar to the HHC of visitors in the baseline phase (22.4%). Using the control leaflet as an intervention increased HHC by 26.4% among children and by 16% among visitors.

Another study [17] explored how exposure to a virtual reality HH promotion video influenced users' HH intention. The results showed a positive message acceptance regarding the participants' attitude towards HH.

A 2-group, comparative observational study [18] assessed the effectiveness of adding new verbal electronic audio reminder (EAR) to a patient HH bundle. In group 1, the participants received a HH bundle with a video, a handout, and a verbal EAR. In the other group, the participants received bundle 2, which including the first two items mentioned above, but not the EAR. Group 1 had a significantly higher mean use of HH agents than group 2, thus showing that adding an EAR to a patient HH bundle improves patient HH.

In a Cluster Randomized Clinical Trial [21] in a geriatric hospital of the University of Geneva Hospitals, Switzerland, all 12 wards and 97 of 306 eligible health care workers (HCWs) volunteered to wear a novel electronic wearable device that delivered real-time feedback on duration of hand rubbing and application of a hand-sized customized volume of alcohol-based handrub (ABHR) and concluded that the use of this device did not change HH compliance, but increased the duration of hand rubbing and volume of ABHR used by HCWs.

### Critical appraisal of the methodological quality

In quasi-randomized study, there were concerns about whether participants in various comparisons were given similar treatment or care apart from the primary exposure or intervention under consideration. Additionally, the presence of a control group and the recurrence of outcome measurements both before and after the intervention or exposure were questionable, compromising the methodological quality.

In clinical trials, there were notable limitations. Questions arose regarding the concealment of allocation to treatment groups. The blinding of participants, those administering the treatments, and outcome assessors to treatment assignments was uncertain. Additionally, the suitability of the trial design and its adherence to or deviation from the standard RCT design, specifically individual randomization and parallel groups, was not clear.

For cross-sectional studies, clarity was lacking in several areas. It was ambiguous whether objective, standard criteria were consistently applied for measuring conditions. The identification of confounding factors and the strategies to address them were also unclear, leading to increased limitations in these sectors.

The studies included in this SR were also critically appraised for their methodological quality using JBI tools (Fig 6).

## Discussion

HH is globally regarded as a crucial strategy for reducing, preventing, and controlling infections, including HAIs, as HH is one of the most important actions for reducing microbial

**Quasi-experimental studies (non-randomized experimental studies)**

| Study | Q1* | Q2† | Q3‡ | Q4§ | Q5‖ | Q6¶ | Q7** | Q8†† | Q9‡‡ |
|---|---|---|---|---|---|---|---|---|---|
| Chen; Chiang, 2007 | Yes | Yes | No | No | No | Yes | Yes | Yes | Yes |
| % | 100 | 100 | 0 | 0 | 0 | 100 | 100 | 100 | 100 |

**Randomised controlled trials**

| Study | Q1§§ | Q2‖‖ | Q3¶¶ | Q4*** | Q5††† | Q6‡‡‡ | Q7§§§ | Q8‖‖‖ | Q9¶¶¶ | Q10**** | Q11†††† | Q12‡‡‡‡ | Q13§§§§ |
|---|---|---|---|---|---|---|---|---|---|---|---|---|---|
| Lary et al., 2020 | Yes | Un‖‖‖ | Yes | No | No | No | Yes | Un‖‖‖ | Yes | Yes | Yes | Yes | Un‖‖‖ |
| Knighton et al., 2018 | Yes | Un‖‖‖ | Yes | Un‖‖‖ | Un‖‖‖ | No | Yes | Yes | Yes | Yes | Yes | Yes | Yes |
| Pires et al., 2021 | Yes | Yes | Un‖‖‖ | No | No | No | Yes | Yes | Yes | Yes | Yes | Yes | Un‖‖‖ |
| % | 100 | 33 | 67 | 0 | 0 | 0 | 100 | 67 | 100 | 100 | 100 | 100 | 33 |

**Cross sectional studies**

| Study | Q1¶¶¶¶ | Q2***** | Q3††††† | Q4‡‡‡‡‡ | Q5§§§§§ | Q6‖‖‖‖‖ | Q7¶¶¶¶¶ | Q8****** |
|---|---|---|---|---|---|---|---|---|
| Boscart et al., 2008 | Yes | Yes | Yes | Un‖‖‖ | Un‖‖‖ | Un‖‖‖ | Yes | Yes |
| Marques et al., 2017 | Yes | Yes | Yes | Un‖‖‖ | Un‖‖‖ | Un‖‖‖ | Yes | Yes |
| Choi; Noh, 2020 | Yes | Yes | Yes | Un‖‖‖ | Un‖‖‖ | Un‖‖‖ | Yes | Yes |
| % | 100 | 100 | 100 | 0 | 0 | 0 | 100 | 100 |

**Fig 6. Critical assessment of methodological quality according to the JBI Critical Appraisal Tool according to the type of study (quasi-experimental studies and non-randomized experimental studies, n = 1; randomized clinical trials, n = 3; cross-sectional studies, n = 3), Ribeirão Preto, SP, Brazil, 2023.** *Q1: Is it clear in the study what is the 'cause' and what is the 'effect' (i.e. there is no confusion about which variable comes first)?; †Q2: Were the participants included in any comparisons similar??; ‡Q3: Were the participants included in any comparisons receiving similar treatment/care, other than the exposure or intervention of interest?; §Q4: Was there a control group?; **Q5: Were there multiple measurements of the outcome both pre and post the intervention/exposure?; ††Q6: Was follow up complete and if not, were differences between groups in terms of their follow up adequately described and analyzed?; ‡‡Q7: Were the outcomes of participants included in any comparisons measured in the same way?; §§Q8: Were outcomes measured in a reliable way?; ***Q9: Was appropriate statistical analysis used?; †††Q1: Was true randomization used for assignment of participants to treatment groups?; ‡‡‡Q2: Was allocation to treatment groups concealed?; §§§Q3: Were treatment groups similar at the baseline?; ****Q4: Were participants blind to treatment assignment?; ††††Q5: Were those delivering the treatment blind to treatment assignment?; ‡‡‡‡Q6: Were outcome assessors blind to treatment assignment?; §§§§Q7: Were treatment groups treated identically other than the intervention of interest?; *****Q8: Was follow up complete and if not, were differences between groups in terms of their follow up adequately described and analyzed?; †††††Q9: Were participants analyzed in the groups to which they were randomized?; ‡‡‡‡‡Q10: Were outcomes measured in the same way for treatment groups?; §§§§§Q11: Were outcomes measured in a reliable way?; ******Q12: Was appropriate statistical analysis used?; ††††††Q13: Was the trial design appropriate and any deviations from the standard RCT design (individual randomization, parallel groups) accounted for in the conduct and analysis of the trial?; ‡‡‡‡‡‡Un: unclear; §§§§§§Q1: Were the criteria for inclusion in the sample clearly defined?; *******Q2: Were the study subjects and the setting described in detail?; †††††††Q3: Was the exposure measured in a valid and reliable way?; ‡‡‡‡‡‡‡Q4: Were objective, standard criteria used for measurement of the condition?; §§§§§§§Q5: Were confounding factors identified?; ********Q6: Were strategies to deal with confounding factors stated?; †††††††Q7: Were the outcomes measured in a valid and reliable way?; ‡‡‡‡‡‡‡‡Q8: Was appropriate statistical analysis used?.

cross-transmission between fomites and patients in health services [2, 7, 22]. Although health services encourage HH compliance and develop several alternatives for implementing this strategy, the literature indicates that these measures are rarely adopted correctly and maintained, and even when adopted, the recommended techniques are not always performed correctly [23, 24]. Accordingly, technologies promoting HH among patients and HCPs are widely valid for ensuring effective and safe healthcare, as shown by the studies included in this review. This study synthesized the evidence related to the use of technologies designed for teaching HH. The used technological resources were different across the studies, in different populations.

Using integrate alcohol gel dispenser system with wearable electronic monitors to emit a signal to perform HH when detect HCP enters or leave protected zones, enable to monitor HCP, and increase compliance [15]. Hospitalized patients have inadequate HH practices that can be improved with HH bundles. The use of electronic hand hygiene monitoring systems has been used during the daily practices of HCP to access their HH compliance and quality [25].

An educational patient HH that incorporates video, handout and verbal electronic audio remember significantly increased HH performance [18]. A blended learning model, which integrate video resources, can be a useful tool for teaching clinical skills as increases knowledge [26].

Other studies used as intervention videos, the authors present a video-based teaching program with higher compliance and accuracy in HH [14]. Another research developed a virtual reality video as a tool to promote HH intention, this improves health promotion as engage preventive behavior [17].

The use of interactive educational intervention technology can increase children's and visitors HH compliance [19]. These data corroborate the findings in a study [8] highlighting the importance of designing technology based on innovative and interactive resources that raise interest in learning while simultaneously enabling a more stimulating and pleasurable learning process.

A gamification solution has been used to collect data on HH and provide feedback on the HH accuracy in nurses [20, 21]. In fact, several studies in the scientific literature have noted that combining interactive technologies with feedback strategies is an important and functional way of implementing a targeted behavior. Concurrently, an integrative review [26] highlighted those interventions using technologies to monitor performance continuously and provide feedback on infection control measures are fundamental after implementing hand hygiene strategies because these methods can help users identify where they are failing and improve their performance.

Overall, the studies analyzed in this review demonstrated that the use of HH educational technologies considerably helps patients, visitors, and relatives learn the procedure and is an efficient tool for improving HHC rates among HCPs. These results corroborate the findings of other studies that aimed at understanding the relationship between the use of technologies and improvement in HH. Developing educational technologies can interactively improve knowledge about HH, in contrast to the standard simple orientation, thereby enhancing compliance in health services, which also improves the quality of care [27].

A study carried out with the purpose of identifying weaknesses and needs regarding HH [1] demonstrated that the use of technical language in educational materials can represent a barrier to understanding the practice, especially for the non-health professional population such as family members and caregivers, and it is urgent to think about educational materials order to serve this population.

Health literacy is also a key component of health education because health literacy determines the ability of an individual to understand and apply health information. Skills such as reading, understanding and analyzing information, following instructions, performing calculations, decoding symbols, and interpreting graphs and diagrams are essential in promoting effective health. Lack of health literacy can have negative health consequences, including lack of understanding of information needed to prevent or treat diseases, failure to adhere to prescribed medical procedures or treatments, lack of understanding of risks associated with risky behaviors, and inability to make informed health decisions [28].

Towards promoting health literacy and enabling health education, new health education technologies have been increasingly used to share information among HCPs and patients. Such technologies, which include electronic devices such as tablets and cell phones, teaching materials in various forms, and virtual reality, are regarded as facilitators of the learning process and healthcare improvement. Using these technologies fosters dialogic and participatory health education, thus raising awareness among patients and professionals beyond the hospital environment. This process occurs through knowledge construction from their own contexts and demands [29].

## Conclusions

Educational technologies implemented in mobile devices (tablets and smartphones), teaching materials, audio visual resources, and virtual reality are the main approaches currently used in teaching professional and lay populations (family members, visitors, relatives, and patients) HH, as shown in the studies included in this review. Based on their findings, educational technologies used to teach HH facilitate the cognitive process, helping to share and improve access of information on how and when to perform HH and opening up opportunities for repetitive learning with reliable and evidence-based content.

The presentation of educational content on HH through technological resources such as mobile devices, virtual reality, and audio-visual reminders, among others, facilitates HH teaching in sharing and accessing information, and helping patients and family members learn the hand washing procedure. Furthermore, they are efficient tools for improving HHC rates among health professionals and open up opportunities for repetitive, efficient, and evidence-based learning.

As a limitation, the small number of studies included in this review had different populations, interventions, and outcomes, due to these it was not possible to perform a metanalysis. Moreover, they have a vulnerable methodological quality, which can compromise generalization of the results to other contexts. However, the results presented in this review may support the development of new studies with other designs on the theme and its practical relevance.

The scientific literature gathered in this review also highlights the importance of encouraging the development of studies aiming to develop technologies for teaching such practice to the entire population. New studies with robust methodological approaches are needed to establish which technological tool for teaching HH is the most effective to be implemented.

## Supporting information

**S1 Fig. PRISMA (Preferred Reporting Items for Systematic Reviews and Meta-Analyses) checklist.**
(DOCX)

## Author Contributions

**Conceptualization:** Daiane Rubinato Fernandes, Carolina Scoqui Guimarães, Renata Cristina de Campos Pereira Silveira.

**Data curation:** Daiane Rubinato Fernandes, Bruna Nogueira dos Santos, Carolina Scoqui Guimarães, Elaine Barros Ferreira, Amanda Salles Margatho, Paula Elaine Diniz dos Reis, Didier Pittet, Renata Cristina de Campos Pereira Silveira.

**Formal analysis:** Daiane Rubinato Fernandes, Bruna Nogueira dos Santos, Elaine Barros Ferreira, Amanda Salles Margatho, Paula Elaine Diniz dos Reis, Didier Pittet, Renata Cristina de Campos Pereira Silveira.

**Funding acquisition:** Daiane Rubinato Fernandes, Renata Cristina de Campos Pereira Silveira.

**Investigation:** Daiane Rubinato Fernandes, Bruna Nogueira dos Santos, Carolina Scoqui Guimarães, Renata Cristina de Campos Pereira Silveira.

**Methodology:** Daiane Rubinato Fernandes, Bruna Nogueira dos Santos, Carolina Scoqui Guimarães, Elaine Barros Ferreira, Amanda Salles Margatho, Paula Elaine Diniz dos Reis, Didier Pittet, Renata Cristina de Campos Pereira Silveira.

**Project administration:** Daiane Rubinato Fernandes, Bruna Nogueira dos Santos, Carolina Scoqui Guimarães, Elaine Barros Ferreira, Amanda Salles Margatho, Renata Cristina de Campos Pereira Silveira.

**Resources:** Daiane Rubinato Fernandes, Renata Cristina de Campos Pereira Silveira.

**Supervision:** Elaine Barros Ferreira, Amanda Salles Margatho, Paula Elaine Diniz dos Reis, Didier Pittet, Renata Cristina de Campos Pereira Silveira.

**Validation:** Daiane Rubinato Fernandes, Bruna Nogueira dos Santos, Carolina Scoqui Guimarães, Elaine Barros Ferreira, Amanda Salles Margatho, Paula Elaine Diniz dos Reis, Didier Pittet, Renata Cristina de Campos Pereira Silveira.

**Visualization:** Daiane Rubinato Fernandes, Bruna Nogueira dos Santos, Carolina Scoqui Guimarães, Elaine Barros Ferreira, Renata Cristina de Campos Pereira Silveira.

**Writing – original draft:** Daiane Rubinato Fernandes, Bruna Nogueira dos Santos, Carolina Scoqui Guimarães, Renata Cristina de Campos Pereira Silveira.

**Writing – review & editing:** Daiane Rubinato Fernandes, Bruna Nogueira dos Santos, Carolina Scoqui Guimarães, Elaine Barros Ferreira, Amanda Salles Margatho, Paula Elaine Diniz dos Reis, Didier Pittet, Renata Cristina de Campos Pereira Silveira.

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
