## [Decision Letter · Decision Letter 0]

10 Oct 2023

PONE-D-23-28730Educational technologies for teaching hand hygiene: systematic reviewPLOS ONE

Dear Dr. Silveira,

Thank you for submitting your manuscript to PLOS ONE. After careful consideration, we feel that it has merit but does not fully meet PLOS ONE’s publication criteria as it currently stands. Therefore, we invite you to submit a revised version of the manuscript that addresses the points raised during the review process.

The manuscript presents a systematic review of Educational technologies for teaching hand hygiene which continues to be a current topic, for health professionals, particularly for students.

The 2 reviewers made very relevant comments that should be carefully considered by the authors.

We look forward to receiving your revised manuscript.

Kind regards,

Maria José Nogueira, Ph.D.

Academic Editor

PLOS ONE

Additional Editor Comments:

The manuscript presents a relevant topic, for health professionals, particularly for students.

Please consider all of the attached reviewers' suggestions and respond to each one carefully.

Reviewers' comments:

Reviewer's Responses to Questions

**Comments to the Author**

1. Is the manuscript technically sound, and do the data support the conclusions?

Reviewer #1: Yes

Reviewer #2: Partly

2. Has the statistical analysis been performed appropriately and rigorously? 

Reviewer #1: Yes

Reviewer #2: I Don't Know

3. Have the authors made all data underlying the findings in their manuscript fully available?

Reviewer #1: Yes

Reviewer #2: Yes

4. Is the manuscript presented in an intelligible fashion and written in standard English?

Reviewer #1: Yes

Reviewer #2: Yes

5. Review Comments to the Author

Reviewer #1: Congratulations on the choice of the topic for the systematic review.

Abstract

Describe the main objetive of the study; Explain how the review was done; Summarize the most important results and their significance; Not exceed 300 words. At this point, it would be importante to clarify which target population was selected for the systematic review, as well as in which care contexts.

Introduction

At this point, it would be importante to demonstrate the worldwide of cross infections caused by improper hand hygiene or incorrect hand hygiene, whith a focus on the risks of mortality and morbidity.

Materials and Methods

1. Clarify what type of descriptors were used for the DeCS/MeSH search in the research.

2. Clarify inclusion and exclusion criteria regarding the population (children, adults, elderly?) and the contexts. If all were included, state this explicitly.

3. Clarify the ethical issues inherent in a systematic review.

Conclusion

Given the richness of the topic, it could be improved by highlighting the conclusions obtained in research of this nature.

It would be important to demonstrate what limitations/difficulties were encountered in this work and what suggestions the authors have for future research.

Reviewer #2: 1) A Systematic Review (SR) of prevalence must include cross-sectional and longitudinal studies. I missed the inclusion of cohort studies to establish a causal relationship between the outcome and the exposure factor (non-hand hygiene). Therefore, for this relationship to be observed, data must be collected at different moments over time;

2) Prevalence SRs, when well designed and rigorously conducted, produce potential guiding and recommendation results for managers and health professionals in the development of specific policies for conditions that most affect and impact a given population. It is necessary for RS authors to be able to identify the different types of prevalence that are being presented in the included studies;

3) Describe in a clear and detailed way the items of the JBI tool that were most important for the analysis of methodological quality. Which questions had the greatest weight when considering the study with greater or lesser methodological quality?;

4) The discussion was approached in a very general way. Relate the findings of the studies included in the SR with the evidence available in the literature.

6. PLOS authors have the option to publish the peer review history of their article (what does this mean?). If published, this will include your full peer review and any attached files.

Reviewer #1: No

Reviewer #2: No

---

## [Author Response · Author response to Decision Letter 0]

30 Oct 2023

Dear Maria José Nogueira, Ph.D.

Academic Editor

PLOS ONE

On behalf of all authors, I inform you that we proceeded with a thorough review of the manuscript based on the reviewers' observations and suggestions. We emphasize that all manuscript changes are marked with yellow highlights. In addition to the reviewers' requests, we also revised Table 6 to present the risk of bias assessment as recommended by JBI. The responses to the comments are described below:

Reviewer #1: 

Congratulations on the choice of the topic for the systematic review.

Abstract

Describe the main objetive of the study; Explain how the review was done; Summarize the most important results and their significance; Not exceed 300 words. At this point, it would be importante to clarify which target population was selected for the systematic review, as well as in which care contexts.

Response: We appreciate and accept the reviewer's suggestion. The main objective of the study was described in the Abstract: “To gather available scientific evidence on technologies used to teach hand hygiene to professional populations and lays involved in health care in the hospital setting”, clarifying the target population and the care context of the systematic review, which was professional populations and lays involved in health care in the hospital setting. We also better explained how the review was done.

Introduction

At this point, it would be importante to demonstrate the worldwide of cross infections caused by improper hand hygiene or incorrect hand hygiene, whith a focus on the risks of mortality and morbidity.

Response: We appreciate and accept the reviewer's suggestion. To accomplish this request, we added the second and third paragraphs of the Introduction section: “The prevention of HAIs should be a priority action for healthcare organizations. According to the WHO, out of 100 patients, seven will acquire at least one healthcare-associated infection during their hospital stay, the risk doubles and may even increase 20 times in low- and middle-income countries. The sicker and more fragile patients become, the greater becomes the risk of HAI and their deathly consequences [3]. Infection prevention and control interventions can reduce infections in healthcare by 70%, specially HH programs can reduce the risk by more than half of dying as a result of infections, and also reduce associated long-term complications and healthcare costs [4].”

Materials and Methods

1. Clarify what type of descriptors were used for the DeCS/MeSH search in the research.

Response: We appreciate and accept the reviewer's suggestion. The DeCS/MeSH descriptors used for the search of this systematic review was described in page 4, and was: “technology”, “health technology” and “hand hygiene”.

2. Clarify inclusion and exclusion criteria regarding the population (children, adults, elderly?) and the contexts. If all were included, state this explicitly.

Response: We appreciate and accept the reviewer's suggestion. Inclusion and exclusion criteria were rewritten on page 5 as: “This review included studies reporting primary, quantitative research findings identifying the use of educational technologies about HH among populations of health professionals, patients, family members and visitors involved in health care in the hospital setting, of any age group, published in English, Portuguese or Spanish, no data limit. Excluded from this review were case reports, case series, secondary studies (other reviews), editorials, letters to the editor, books, book chapters, guidelines, expert opinion articles, experience reports, conference proceedings and abstracts, dissertations and theses, in addition to studies outside the scope of this review”, clarifying that the population was health professionals, patients, family members and visitors, of any age group, in the context of health care in the hospital setting.

3. Clarify the ethical issues inherent in a systematic review.

Response: We appreciate and accept the reviewer's suggestion. Since the review did not involve human participants, it did not require ethical committee approval. However, the authors conducted this review with a strong emphasis on transparency and reproducibility, avoiding bias, and ensuring a comprehensive and fair evaluation. This was clarified on page 6.

Conclusion

Given the richness of the topic, it could be improved by highlighting the conclusions obtained in research of this nature. It would be important to demonstrate what limitations/difficulties were encountered in this work and what suggestions the authors have for future research.

Response: We appreciate and accept the reviewer's suggestion. We highlighted the conclusions obtained in this systematic review adding the paragraph: “The presentation of educational content on HH through technological resources such as mobile devices, virtual reality, and audio-visual reminders, among others, facilitates HH teaching in sharing and accessing information, and helping patients and family members learn the hand washing procedure. Furthermore, they are efficient tools for improving HHC rates among health professionals and open up opportunities for repetitive, efficient, and evidence-based learning”. 

We demonstrated the limitations/difficulties encountered in this work with the paragraph: “As a limitation, the small number of studies included in this review had different populations, interventions, and outcomes, due to these it was not possible to perform a metanalysis. Moreover, they have a vulnerable methodological quality, which can compromise generalization of the results to other contexts. However, the results presented in this review may support the development of new studies with other designs on the theme and its practical relevance.”

We added the suggestions for future research with the paragraph: “The scientific literature gathered in this review also highlights the importance of encouraging the development of studies aiming to develop technologies for teaching such practice to the entire population. New studies with robust methodological approaches are needed to establish which technological tool for teaching HH is the most effective to be implemented”.

Reviewer #2: 

1) A Systematic Review (SR) of prevalence must include cross-sectional and longitudinal studies. I missed the inclusion of cohort studies to establish a causal relationship between the outcome and the exposure factor (non-hand hygiene). Therefore, for this relationship to be observed, data must be collected at different moments over time.

Response: Thank you for your comment. We did not plan to conduct a prevalence review, as this review did not assess how many people did HH (yes- hand hygiene) and how many people did not (non-hand hygiene). Therefore, quasi-experimental studies and randomized clinical trials, were included. The studies addressed the use of technologies to teach professionals or lay populations (such as family members, visitors, relatives, and patients) about hand hygiene and to support and encourage their development for this purpose, as proposed in the objective of our review. Finally, given the heterogeneity of the included studies, we did not perform a meta-analysis of the obtained data.

2) Prevalence SRs, when well designed and rigorously conducted, produce potential guiding and recommendation results for managers and health professionals in the development of specific policies for conditions that most affect and impact a given population. It is necessary for RS authors to be able to identify the different types of prevalence that are being presented in the included studies.

Response: Thank you for your comment. In this review, we did not intend to conduct a prevalence review with meta-analysis. While the topic of hand hygiene is broad and comprehensive, the use of technologies for teaching hand hygiene is still in its early stages, which would preclude analyses like prevalence meta-analyses. For this reason, we adhered to the objective of our review and included studies that addressed educational strategies for teaching hand hygiene, presenting the results in a descriptive manner.

3) Describe in a clear and detailed way the items of the JBI tool that were most important for the analysis of methodological quality. Which questions had the greatest weight when considering the study with greater or lesser methodological quality?

Response: We appreciate and accept the reviewer's suggestion. To accomplish this request, we added three paragraphs below the subitem “Critical appraisal of the methodological quality” on page 10.

4) The discussion was approached in a very general way. Relate the findings of the studies included in the SR with the evidence available in the literature.

Response: We appreciate and accept the reviewer's suggestion. To accomplish this request, we added and rewrite some paragraphs of the Discussion section, corroborating our findings with data from the literature.

Renata Cristina de Campos Pereira Silveira

RN PhD Associate Professor

University of São Paulo

Ribeirão Preto College of Nursing

---

## [Editor Report · Decision Letter 1]

7 Nov 2023

Educational technologies for teaching hand hygiene: systematic review

PONE-D-23-28730R1

Dear Dr. Renata Cristina de Campos Pereira Silveira,

We’re pleased to inform you that your manuscript has been judged scientifically suitable for publication and will be formally accepted for publication once it meets all outstanding technical requirements.

Kind regards,

Maria José Nogueira, Ph.D.

Academic Editor

PLOS ONE

Additional Editor Comments (optional):

Dear Autor

All the concerns mentioned by the reviewers, were clarified by the authors, which provided greater clarity and robustness to the manuscript.

Now the work complies with the journal's scientific requirements.

The manuscript may be accepted for publication.
---

## [Editor Report · Acceptance letter]

4 Jan 2024

PONE-D-23-28730R1 

PLOS ONE

Dear Dr. Silveira, 

I'm pleased to inform you that your manuscript has been deemed suitable for publication in PLOS ONE. Congratulations! Your manuscript is now being handed over to our production team.

Kind regards, 

on behalf of

Professor Maria José Nogueira 

Academic Editor

PLOS ONE